# Anti-Müllerian Hormone, Growth Hormone, and Insulin-Like Growth Factor 1 Modulate the Migratory and Secretory Patterns of GnRH Neurons

**DOI:** 10.3390/ijms22052445

**Published:** 2021-02-28

**Authors:** Rossella Cannarella, Alyssa J. J. Paganoni, Stefania Cicolari, Roberto Oleari, Rosita A. Condorelli, Sandro La Vignera, Anna Cariboni, Aldo E. Calogero, Paolo Magni

**Affiliations:** 1Department of Clinical and Experimental Medicine, University of Catania, 95123 Catania, Italy; rosita.condorelli@unict.it (R.A.C.); sandrolavignera@unict.it (S.L.V.); acaloger@unict.it (A.E.C.); 2Department of Pharmacological and Biomolecular Sciences, Università degli Studi di Milano, 20133 Milan, Italy; alyssa.paganoni@unimi.it (A.J.J.P.); stefania.cicolari@unimi.it (S.C.); roberto.oleari@unimi.it (R.O.); paolo.magni@unimi.it (P.M.); 3IRCCS MultiMedica, Sesto S. Giovanni, 20099 Milan, Italy

**Keywords:** hypogonadotropic hypogonadism, GnRH, AMH, GH, IGF1, neuron migration, GnRH secretion

## Abstract

Anti-Müllerian hormone (AMH) is secreted by Sertoli or granulosa cells. Recent evidence suggests that AMH may play a role in the pathogenesis of hypogonadotropic hypogonadism (HH) and that its serum levels could help to discriminate HH from delayed puberty. Moreover, the growth hormone (GH)/insulin-like growth factor 1 (IGF1) system may be involved in the function of gonadotropin-releasing hormone (GnRH) neurons, as delayed puberty is commonly found in patients with GH deficiency (GHD) or with Laron syndrome, a genetic form of GH resistance. The comprehension of the stimuli enhancing the migration and secretory activity of GnRH neurons might shed light on the causes of delay of puberty or HH. With these premises, we aimed to better clarify the role of the AMH, GH, and IGF1 on GnRH neuron migration and GnRH secretion, by taking advantage of previously established models of immature (GN11 cell line) and mature (GT1-7 cell line) GnRH neurons. Expression of *Amhr*, *Ghr*, and *Igf1r* genes was confirmed in both cell lines. Cells were then incubated with increasing concentrations of AMH (1.5–150 ng/mL), GH (3–1000 ng/mL), or IGF1 (1.5–150 ng/mL). All hormones were able to support GN11 cell chemomigration. AMH, GH, and IGF1 significantly stimulated GnRH secretion by GT1-7 cells after a 90-min incubation. To the best of our knowledge, this is the first study investigating the direct effects of GH and IGF1 in GnRH neuron migration and of GH in the GnRH secreting pattern. Taken together with previous basic and clinical studies, these findings may provide explanatory mechanisms for data, suggesting that AMH and the GH-IGF1 system play a role in HH or the onset of puberty.

## 1. Introduction

Delayed puberty is defined in males as lack of testicular volume enlargement at 14 years of age and, in females, as the absence of thelarche at 13 years (2–2.5 standard deviations above mean). It is a common condition, affecting 2% of children at pubertal age [1]. The causes of delayed puberty are often unrecognized [1] and may be due to a lack of activation of the gonadotropin-releasing hormone (GnRH) neurons. Similarly, hypogonadotropic hypogonadism (HH) represents a congenital or acquired disease often associated with abnormal function of GnRH neurons [1].

Understanding the stimuli that promote GnRH neuron migration and secretion could shed light on the causes of delayed puberty or HH, possibly offering therapeutic choices. In this context, scanty evidence assigns anti-Müllerian hormone (AMH) and the growth hormone (GH)/insulin-like growth factor 1 (IGF1) system a role in GnRH neuron function.

AMH, also known as Müllerian-inhibiting substance (MIS) or factor (MIF), is a 560-amino acid polypeptide whose *C*-terminal domain displays a high homology with human transforming growth factor β (TGFβ) and porcine inhibin B. It is encoded by a gene made of 5 exons and maps in the 19p13.3 chromosome [2]. The AMH receptor (AMHR or AMHR2) is a 573-amino acid serine-threonine kinase with a single transmembrane domain belonging to the family of type II receptors for TGFβ-related proteins. In males, AMH is secreted by testicular Sertoli cells and it is required for male differentiation during fetal life [3].

Recent studies have explored the role of AMH in GnRH neuron firing. In greater detail, AMHR has been found in GnRH neurons from three human fetuses at the ninth week of gestation (in this phase, they are still located in the nasal region, at the beginning of their migratory path), in the adult hypothalamus (the post-mortem analysis was carried out in a man and in women), and in mice by immunostaining [4]. Stimulation with recombinant AMH (rAMH), at the doses of 0.04, 0.4, and 4 nM (equivalent to 1, 10, and 100 ng/mL) (~0.1 nM is the physiologic concentration in this animal model), increases GnRH neuron firing and GnRH secretion (the dose of 125 nM was used for this outcome) in 4–8-month-old female mice [4]. Besides, the intracerebroventricular administration of AMH led to luteinizing-hormone (LH) secretion at the doses of 0.5, 1, and 3 nM, in mice [4].

AMHRs have also been found in the anterior pituitary of post-pubertal heifers. Incubation of anterior pituitary cells with increasing doses of AMH (0, 1, 10, 100, or 1000 pg/mL) for 3.5 days stimulated basal LH and follicle-stimulating hormone (FSH) secretion [5]. Finally, *AMHR* expression is modulated by GnRH in gonadotropic cells, via Early growth response protein 1 (Egr1) and Forkhead box protein O1 (FOXO1) [6]. Taken together, these data seem to support the role of AMH in the stimulation of GnRH neurons, but the evidence is still limited and the real importance of AMH in post-natal life is not entirely clear.

Few data are currently available on the influence of the GH/IGF1 axis on the function of GnRH neurons. GH is a 22 kDa protein, released in pulses by the somatotropic cells of the anterior pituitary gland, under the dual control of hypothalamic GH-releasing hormone and somatostatin. GH binds to GH receptors (GHRs) which are expressed in many organs and tissues, including the liver and gonads. In the first, GH stimulates the synthesis of IGF1, which promotes body growth [7]. The presence of “acromegalic” levels of IGF1 in the blood of healthy children during puberty has led to the hypothesis that the GH–IGF1 axis may be important for successful and complete and pubertal development [5]. Accordingly, GnRH neurons express IGF1 receptors (IGF1Rs) in both male and female mice [8]. Furthermore, an in vitro study showed that IGF1 at the concentration of 10 ng/mL increases GnRH secretion by 80–100% in GT1-7 cells after a 2-h incubation but decreases the release of this neuropeptide after 6 and 24 h of incubation [9]. These results were confirmed by another study in GT1-7 neurons cultured in vitro, which showed an increase in GnRH release (+139%) after 2 h of incubation with IGF1 (10 ng/mL) and a decrease from the fourth hour of incubation (48%) up to 48 h (60%). These results suggest that IGF1 stimulates the release of GnRH stored in the vesicles, rather than its biosynthesis [10]. Finally, a previous study reported that blocking the IGF1 signaling pathway affects the spatial organization of GnRH neurons in zebrafish [11], suggesting that IGF1 might influence the migration of GnRH neurons.

Despite this evidence, no data are currently available on the influence of the GH/IGF1 system on GnRH neuron migration and secreting patterns.

To further evaluate the role of AMH, GH, and IGF1 on GnRH neuron migration and GnRH secretion, we exploited established [12,13,14] models of immature (GN11 cell line) and mature (GT1-7 cell line) GnRH neurons. The expression of *Amhr*, *Ghr*, and *Igf1r* genes was evaluated in both cell lines. The cells were then incubated with increasing concentrations of AMH, GH, or IGF1 and the migration pattern of GN11 cells and GnRH secretion by GT1-7 cells were assessed.

## 2. Results

### 2.1. Immortalized GnRH Neurons (GN11 and GT1-7 Cell Lines) Express Different mRNA Levels for Amhr2, Ghr, and Igf1r

We first evaluated the transcript levels *of Amhr2, Ghr*, and *Igf1r* genes, encoding for AMH, GH, and IGF1 receptors, in GN11 and GT1-7 cells, which represent models of immature migrating and maturing secreting GnRH neurons, respectively. The expression of *Amhr2, Ghr*, and *Igf1r* genes, evaluated by qPCR, was found to be developmentally regulated, with greater levels in mature GT1-7 cells compared to immature GN11 cells (Figure 1A), for all the three genes analyzed. As expected, GnRH gene expression, used as a control, was found to be lower in GN11 cells compared to GT1-7 cells (Figure 1B).

### 2.2. Effect of Treatment with AMH, GH, and IGF1 on GN11 Cell Migration

To study the effects of AMH, GH, and IGF1 on the migration of GnRH neurons, we employed a chemotaxis assay and GN11 cells as a model of GnRH neurons with migratory ability. As shown in Figure 2, GN11 cell chemomigration was stimulated in a dose-dependent manner by the presence of AMH in the lower compartment. Specifically, we found that AMH, at the concentration of 150 ng/mL, was able to stimulate a chemomigratory response similar to 10% FBS, a well-known potent chemomigratory stimulator used as a positive control.

We then assessed, following a similar experimental approach, the effects of GH on GN11-chemomigration. Although to a lower extent compared to AMH, GN11-chemomigration was also stimulated by the presence of GH, with a dose–response effect (Figure 3). Specifically, we found a significant increase in the number of migrated GN11 cells at the doses of 300 and 1000 ng/mL compared to DMEM alone, used as a negative control.

Finally, we also tested the effect of IGF1 on the chemomigration of GN11 cells and found that IGF1 was able to significantly increase the number of migrated GN11 cells compared to DMEM at the doses of 30 and 100 ng/mL. Interestingly, we found that, at the concentration of 100 ng/mL, IGF1 was able to induce a very potent chemomigratory response, significantly greater than that induced by 10% FBS, strongly suggesting a role of IGF1 in the migration of GnRH neurons (Figure 4).

### 2.3. Effects of AMH, GH, and IGF1 on GN11 Cell Morphology

Cytoskeleton rearrangements are essential during cell migration. Thus, to visualize the possible effects of AMH, GH, and IGF1 on cytoskeleton rearrangements in GN11 cells and to confirm in situ the effects observed on migration, we stained GN11 cells for actin and tubulin after incubation for 2 h with the two most effective concentrations of the three hormones used in the chemomigration assays. As shown in Figure 5, Figure 6 and Figure 7, both concentrations of AMH, GH, and IGF1, which were found able to stimulate GN11 chemomigration, induced, although to a different extent, morphological changes consistent with an induction of cell movement. Specifically, upon treatment, GN11 cell bodies were more elongated, as is typically observed in migrating neurons. The percentage of cells with an elongated shape for the different treatments is reported in Table 1.

### 2.4. Effect of Treatment with AMH, GH, and IGF1 on GnRH Release by GT1-7 Cells

The effect of a 90-min exposure of GT1-7 cells to AMH, GH, and IGF1 on GnRH release and accumulation into the medium was then assessed. Forskolin (10 mg/mL) treatment was used as the positive control. Treatment with different concentrations (range: 1.5–150 ng/mL) of AMH increased GnRH release into the medium, reaching significant values (*p* < 0.05) at 1.5 ng/mL (+22%) and 150 ng/mL (+25%) (Figure 8).

Exposure of GT1-7 cells to increasing concentrations of GH (range: 3–1000 ng/mL) moderately (maximum +22%), but significantly (*p* < 0.001 or <0.0001) stimulated GnRH release at 30–300 ng/mL (Figure 9). No effect was observed at the concentration of 3 and 10 ng/mL as well as 1000 ng/mL (not shown).

GnRH release by GT1-7 cells was also moderately, but significantly (*p* < 0.05 or <0.01), stimulated by 3 and 10 ng/mL IGF1, but not at lower (0.3 and 1 ng/mL) and higher (30 and 100 ng/mL) concentrations (Figure 10).

## 3. Discussion

In the present study, the effects of AMH, GH, and IGF1 on GnRH neuron migration and secretion pattern were assessed, by taking advantage of models of immature (GN11 cell lines) and mature (GT1-7 cell lines) GnRH neurons, that were validated in previous studies [12,13,14]. First, we investigated the expression of the *Amhr2*, *Ghr*, and *Igf1r* genes, which were all detectable in both GN11 and GT1-7 cell lines although, expectably, at higher levels in mature GnRH neurons (GT1-7 cell lines). The effects of graded concentrations of AMH, GH, and IGF1 were then evaluated in both cell lines.

AMH, GH, and IGF1 significantly enhanced GN11 cell migration in a dose-dependent manner and, in line with this, they all promoted cytoskeletal rearrangements that were consistent with cell migration. Incubation with AMH, GH, or IGF1 for 90 min resulted in a significant increase of GnRH secretion by GT1-7 cells. The concentrations of AMH, GH, and IGF1 used in the present study were chosen according to the available evidence. Particularly, the study by Cimino and colleagues [4], who employed 1, 10, and 100 ng/mL AMH for intraventricular injections, was used as a reference point, and therefore, the range 1.5–150 ng/mL was selected to perform the experiments. Concerning IGF1, the already explored concentrations in GT1-7 cell lines include 1, 10, and 100 ng/mL [9] and 0.1, 10 and 100 ng/mL [10]. A concentration of 0.1 or 10 ng/mL has been previously used, although in GT1, GT1-1, or NLT cell lines [15,16,17]. Thus, the dose range of 1.5–150 ng/mL was chosen for the incubations. To the best of our knowledge, no evidence has been published so far on the effects of GH on GnRH neurons. The dose of 125 ng/mL has been used in cultured adipocytes to test the effects of GH on lipid accumulation and cell maturation [18]. Due to the lack of data, we selected a wide dose range (3–1000 ng/mL) to perform the experiments.

As regards the time of incubation, we took advantage of a GT1-7 cell line, which we were known to secrete GnRH already after 30–60 min of incubation with leptin [12]. The experiments performed by Malone and coworkers on GT1-7 reported that the secretion of GnRH was present after the 4th hour of incubation with AMH [19]. Worryingly, time-course experiments and results at earlier times of incubation were not shown [19]. Another study selected the times of 2, 4, 8, 24, and 48 h [10] to investigate the effects of IGF1 incubation on GnRH secretion in GT1-7 cell lines, reporting an increase (+139%) at the 2nd hour, but a decrease at the subsequent times [10]. Based on these data, a time of incubation of 90 min was chosen to perform the experiments.

The findings of the present study suggest a role for AMH, GH, and IGF1 on GnRH neuron migration, during fetal life, from olfactory placode to hypothalamus-preoptic area, and for AMH and GH in the induction of GnRH secretion in the post-natal life.

AMHRs have already been reported in GnRH neurons from human fetuses and in adult human [4], mouse [4], and post-pubertal heifers hypothalamic [5]. Consistent with our findings, loss-of-function heterozygous mutations of the *AMH* or *AMHR2* genes have been recently described in the 3% of probands with congenital HH (cHH) [19]. Importantly, in vitro experiments functionally validated the disease-causing role of both these gene mutations. GN11 cells transfected with the *AMH* or *AMHR2* gene variants were indeed unable to migrate after incubation with AMH compared to the wild-type (WT) cell lines. Similarly, transfected GT1-7 cells were able to secrete insignificantly higher amounts of GnRH compared to the WT cells after incubation with AMH (50 ng/mL) [19]. Furthermore, the same authors reported an abnormal development of the peripheral olfactory system and defective embryonic migration in *Amhr2* deficient mice [19]. Taken together, these data suggest the importance of AMH and its receptor for the proper development and function of GnRH neurons in humans since their loss-of-function mutation can lead to cHH [19].

Several lines of other evidence, mostly coming from the animal model, further confirmed this theory. Cimino and colleagues [4] showed the involvement of AMH in the GnRH neuron cell firing. Indeed, the intraventricular injection of rAMH resulted in GnRH increase and, in turn, in LH secretion in mice [4]. The enhancing effect of LH secretion following in vitro incubation with AMH has been confirmed also elsewhere [5].

AMH is secreted by immature and proliferating Sertoli cells starting from the 6th week of gestation. Its levels remain high during the 2nd and the 3rd trimesters [1]. Its physiological role is well-known in the pre-natal lifetime. AMH is indeed involved in the involution of Müllerian ducts, which would otherwise differentiate into the uterus and fallopian tubes. Accordingly, *AMH* or *AMHR* gene mutations lead to type I or type II persistent Müllerian ducts syndrome (PMDS), respectively, which is included among the disorders of sex development [20]. Notably, AMH levels increase in the first year of life, remaining high until puberty. When puberty starts, Sertoli cells, now expressing the androgen receptor [21], transit from a proliferative and immature state to a quiescent and mature stage and the levels of AMH decrease [22].

In the female gender, AMH is secreted by granulosa cells in post-natal life and its serum levels highly correlate with the ovarian reserve [23,24].

The physiological role of AMH and its receptor in post-natal life and, particularly, from birth to puberty, is currently unknown. The presence of AMHR2 outside the gonads, and, particularly, in gonadotropic cells of the anterior pituitary gland and the hypothalamus, suggests a role for AMH as a neuroendocrine regulation of the hypothalamic-pituitary-gonadal function [25]. Interestingly, clinical evidence indicates that AMH serum levels may help in differentiating the patients with HH from those with delayed puberty in childhood [26,27,28]. Accordingly, discriminating between these two conditions is very challenging in the pre-pubertal phase since sex hormones and gonadotropins serum levels are still not indicative. In this contest, AMH levels seem lower in male children with HH than peers with delayed puberty [26,27,28]. Hypothetically, the lower levels of AMH may negatively influence the function of the GnRH neurons, thus leading to the onset of HH. Therefore, the results of the present study, which are in line with previously published data [4,19], may provide further explanatory mechanisms to better understand clinical evidence [26,27,28]. This, in turn, strengthens the possible role of AMH as a biomarker of HH in pre-pubertal life. Moreover, given its enhancing effect on GnRH secretion, targeted studies might be designed to investigate the possible therapeutic properties of AMH in HH.

As previously introduced, scanty data are currently available on the role of the GH-IGF1 system on GnRH neuron migration and secretion. In deeper details, there is no evidence of direct effects of GH on GnRH neurons. To the best of our knowledge, this study is the first to report the stimulatory effect of GH on GN11 cell migration, and on GnRH secretion from GT1-7 cells.

Indirect evidence indicates that IGF1 may be involved in the migration of GnRH neurons since the blockage of the IGF1 signaling pathway can influence their spatial organization in zebrafish [11]. Accordingly, our experiments confirm the enhancing effect of IGF1 on GN11 cell migration.

Moreover, we observed a moderate but significant stimulatory effect of IGF1 on GnRH secretion, after 90 min of incubation. Other authors have previously investigated the effects of this hormone on GnRH secretion in GT1-7 cell lines. Particularly, Longo and collaborators have reported an increase after 2 h of incubation, although it was followed by a decrease after prolonging the length of incubation [10]. Indeed, GnRH secretion showed a decrease starting from the 4th hour of incubation (48%), and until 48 h (60%) [10]. Therefore, the findings of the present study, framed in the context of previous literature, might indicate that IGF1 may enhance the release of the GnRH stored in vesicles, more than its synthesis.

The serum levels of IGF1 physiologically increase in healthy children across puberty [7]. Delayed puberty is a common finding in children with GH deficiency (GHD) [29] or patients with Laron syndrome, a genetic form of GH resistance with undetectable or very low IGF1 levels [30,31,32]. This suggests that IGF1 serum levels within the normal range are important for the timely onset of puberty.

Interestingly, IGF1 serum levels are lower in children with HH than in those with delayed puberty [28], possibly indicating a role of the GH-IGF1 axis in HH. Supporting this finding, novel variants of the *IGF1* gene have been described among a cohort of 138 patients with idiopathic HH. However, to confirm the role of these variants in the pathogenesis of HH, a functional validation by in vitro models is needed [33].

Therefore, the results of the present study can provide basic evidence supporting the role of the GH-IGF1 system in the onset of puberty, as already hypothesized [7]. If further confirmed, this knowledge can have both diagnostic and therapeutic value in the clinical practice. More in detail, serum AMH levels might be a reliable marker to differentiate constitutional puberty delay from HH in developmental age. Similarly, borderline-low IGF1 levels may also negatively influence GnRH neuron firing, thus leading to delayed puberty. Concerning the possible therapeutic implications, no study has investigated the role of AMH as a therapeutic agent for the treatment of HH or puberty delay, probably because the scientific evidence of its influence on GnRH neurons has been produced only recently. In vivo animal studies should be developed to specifically assess this issue.

Treatment with IGF1 is very rarely prescribed, being reserved for patients with GH resistance. In contrast, GH is therapeutically available and its prescription is allowed for the treatment of patients with GHD. Based on the results of the present study, GH deserves to be investigated in cases of non-GHD-related puberty delay or HH. Proper therapeutic schemes are to be tested to avoid the negative consequences on the glycolipid metabolism. Therefore, animal studies designed to evaluate the role of GH for the treatment of non-GHD-related puberty delay or HH should be carried out.

Finally, this study has been primarily performed with immortalized GnRH neuron cell lines. The lack of in vivo data represents a limit of the present study, which should prompt to design of further experiments focused to confirm the expression of AMHR2, GHR, and IGF1R in GnRH neurons also in vivo.

In conclusion, this study shows that AMH, GH, and IGF1 support GN11 cell chemomigration and can directly stimulate GnRH secretion in GT1-7 cells. Moreover, the receptors for all these hormones are expressed in these cell lines. Taken together with previous basic and clinical data, these findings may provide explanatory mechanisms for the clinical data addressing AMH and the role of the GH-IGF1 system in HH or the onset of puberty.

## 4. Materials and Methods

### 4.1. Chemicals

Human recombinant AMH and GH, and mouse recombinant IGF1, were purchased from R&D Systems, Inc. (Minneapolis, MN, USA). Forskolin and the other analytical reagents were purchased from Merck (Milan, Italy). GH and IGF1 were reconstituted in PBS, while AMH in 4 mM HCl with 0.1% bovine serum albumin (Euroclone, Milan, Italy), according to manufacturer’s instructions.

### 4.2. Cell Cultures

GN11 cells (a kind gift of S. Radovick, Children’s Hospital, Division of Endocrinology, Boston, MA, USA) and GT1-7 cells (a kind gift of R. I. Weiner, San Francisco, CA, USA) were grown in a monolayer at 37 °C in a humified CO_2_ incubator. Cells were cultured in Dulbecco’s minimum essential medium (DMEM) supplemented with 1 mmol/L sodium pyruvate, 100 µg/mL streptomycin, 100 U/mL penicillin, 2 mmol/L glutamine (Euroclone, Milan, Italy), and 10% fetal bovine serum (FBS) (Euroclone, Milan, Italy). The medium was replaced at 3-day intervals. Sub-confluent cells were harvested with 0.05% trypsin/0.02% ethylenediaminetetraacetic acid (EDTA) (Euroclone, Milan, Italy). After collection, samples were snap-frozen in liquid nitrogen and stored at −80 °C until RNA extraction or other analyses.

### 4.3. RNA Extraction and Gene Expression Analyses

The total RNA extracted from GN11 and GT1-7 cells was collected using the Trizol-chloroform method [34]. After extraction, 1 μg of total RNA was reverse transcribed with random hexamers and MultiScribe reverse transcriptase (Applied Biosystems, Monza, Italy) according to the manufacturer’s instructions. mRNA levels of murine *Gnrh1* (fw 5′-CGTTCACCCCTCAGGGATCT-3′; rev 5′-CTCTTCAATCAGACTTTCCAGAGC-3′), *Amhr2* (fw 5′-CCTGGGAATGTTTCTCGTGT; rev 5′-TGGATTACCTGGGAGAAACG-3′), *Igf1r* (fw 5′-TGAACCCCGAGTATTTCAGC-3′; rev 5′-CACTCTGGTTTCGGGTTCAT-3′), and *Ghr* (fw 5′-GTTCCCCTGAACTGGAGACA-3′; rev 5′-AGGGCATTCTTTCCATTCCT-3′) genes were quantified by qPCR on a Biorad CFX Connect thermal cycler with Luna Universal qPCR Master Mix (NEB, Ipswich, MA, USA) in 10 μL reactions, with a final concentration of 0.25 μM for each primer. The cycling conditions were 95 °C for 1 min, followed by 40 cycles of 15 s at 95 °C, 30 s at 60 °C, and 30 s at 72 °C. A final melting curve analysis assured the authenticity of the target product. Triplicate samples were run in all reactions; first-strand DNA synthesis reactions without reverse transcriptase were used as controls. The ΔCq value and the ΔΔCq were calculated relative to control samples using quantification cycle (Cq) threshold values that were normalized to the housekeeping gene, Gapdh (fw: 5′-CATCCCAGAGCTGAACG-3′; rev 5′-CTGGTCCTCAGTGTAGCC-3′).

### 4.4. Chemomigration Assays

Migration studies were performed using a 48-well Boyden’s micro-chemotaxis chamber, according to manufacturer’s instructions (Neuro Probe, Gaithersburg, MD, USA) and as described previously [14]. Briefly, cells, grown in the regular complete medium until sub-confluence, were collected with 0.05% trypsin/0.02% EDTA, and the cell suspension (0.10 × 10^6^ cells/50 µL DMEM/0.1% BSA) was transferred in the open-bottom wells of the upper compartment. The two compartments were separated by a polyvinyl-pyrrolidone-free polycarbonate porous membrane (8 µm pores) precoated with gelatin (0.2 mg/mL in PBS) (DBA Italia, Milan, Italy). Twenty-eight microliters of the control experimental medium (DMEM) or control chemoattractant (10% FBS) or the test compounds at different concentrations were placed into the wells of the lower compartment of the chamber. The chamber was then kept in the cell culture incubator at 37 °C for 3 h. At the end of the incubation period, the cells migrated through the pores and adherent to the lower side of the membrane were fixed with 100% MeOH, stained using the Diff-Quik kit (Biomap, Milan, Italy), and mounted onto glass slides. For quantitative analysis, cells were counted using a 20× objective (Zeiss Plan-Neofluar 20×, NA 0.50) on a light microscope. Three random objective fields of stained cells were counted for each well with ImageJ32 Software v1.46 (NIH, Bethesda, MD, USA), and the mean number of migrating cells/field was calculated.

### 4.5. Cytoskeleton Labeling

For cytoskeleton labeling, GN11 cells were seeded at a density of 5 × 10^3^ cells/well in a 24-multiwell plate, two days after they were treated for 60/120 min with the different hormones and fixed with 4% paraformaldehyde-PBS for 15 min at RT and immunostained as previously described [35]. Briefly, fixed cells were then incubated with an anti-alpha-tubulin primary antibody at room temperature for 2 h, followed by an anti-mouse 488-conjugated secondary antibody at room temperature for 2 h. To detect F-actin, cells were treated with phalloidin-TRITC (1:400, in PBS) for 30 min at 37 °C before mounting with Mowiol. Preparations were then examined with an epifluorescent fluorescent microscope (Zeiss, Milano, Italy), and images were acquired with a Zeiss LSM900 laser scanning confocal microscope and a 40× objective (Zeiss Plan-Apochromat 40×, NA 1.3, Oil-immersion). ZEN 3.0 software (Zeiss, Milano, Italy) was used to process z-stacks at 0.25 µm intervals and generate maximal intensity projection images. Quantification of the percentage of the cells with an “elongated” shape was performed by counting 5 fields/conditions by ImageJ.

### 4.6. GnRH Secretion Assay

GT1-7 cells were seeded in 24-well plates (150,000 cells/well). After 24 h, cells were incubated for 90 min with an experimental medium containing different concentrations of AMH, GH, and IGF1. The 90-min time frame was selected by preliminary experiments as the shortest incubation time allowing to appreciate a variation of GnRH levels in the conditioned medium, but avoiding significant degradation of the peptide [12]. At the end of the treatment, the medium was collected and stored at −20 °C. Subsequently, the concentration of GnRH in the media was determined by an EIA KIT provided by Phoenix Pharmaceuticals (#EK-040-02CE; Karlsruhe, Germany) (sensitivity 0.08 ng/mL, intraassay variation <10%, interassay variation <15%). The protocol was performed according to the manufacturer’s guidelines. The experiments were repeated at least three times.

### 4.7. Analysis of the Data

The statistical analysis was conducted using the Prism 8.2.1 statistical analysis package (GraphPad Software, San Diego, CA, USA). Data are expressed as mean ± standard deviation (SD). Differences between treatment groups were evaluated by two-tailed unpaired Student’s *t*-test or one-way ANOVA, followed by post-hoc Dunnett’s test and considered significant at *p* < 0.05.

## Figures and Tables

**Figure 1 ijms-22-02445-f001:**
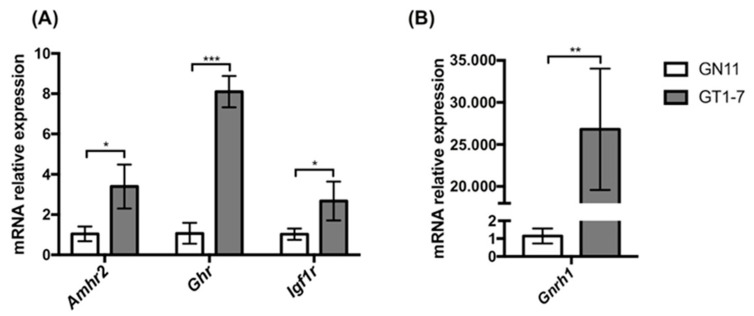
Analysis of the expression of the *Amhr2*, *Ghr*, and *Igf1r* (**A**) and *Gnrh1* (**B**) genes in GN11 and GT1-7 cells. Data are expressed as mean ± SD, *n* = 3. * *p* < 0.05; ** *p* < 0.01; *** *p* < 0.001 (Two-tailed unpaired Student’s *t*-test). *Amhr2*, *Anti-Müllerian hormone receptor 2*; *Ghr*, *growth hormone receptor*; *Igf1r*, *Insulin-like growth factor 1 receptor*.

**Figure 2 ijms-22-02445-f002:**
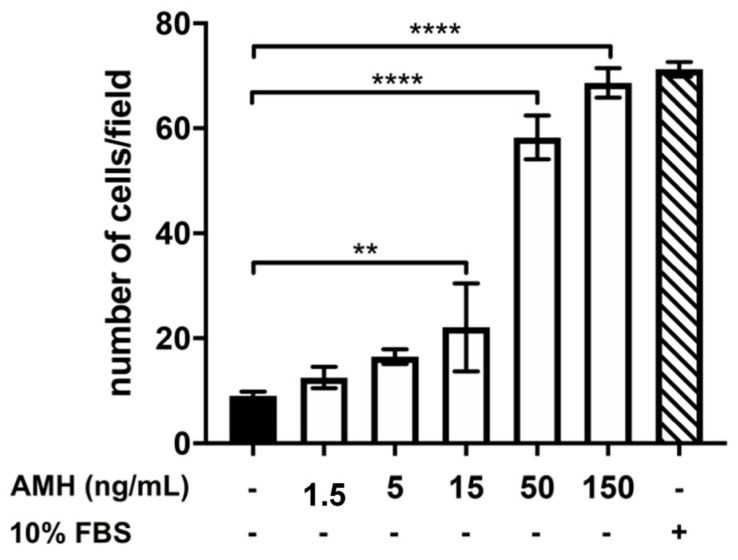
Effect of different concentrations of anti-Müllerian hormone (AMH) on GN11 cell migration. Data are expressed as mean ± SD, *n* = 8. One experiment representative of three separate experiments is shown. ** *p* < 0.01; **** *p* < 0.0001 (One-way ANOVA followed by the Dunnett post-hoc test).

**Figure 3 ijms-22-02445-f003:**
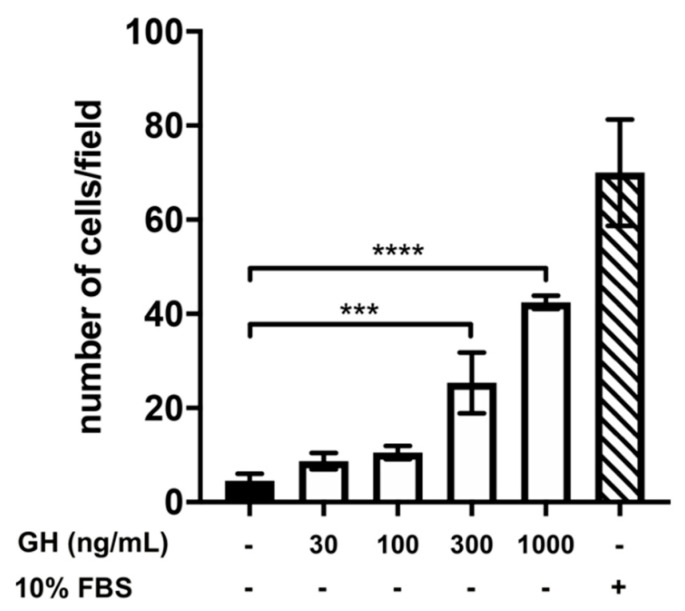
Effect of different concentrations of growth hormone (GH) on GN11 cell migration. Data are expressed as mean ± SD, *n* = 8. One experiment representative of three separate experiments is shown. *** *p* < 0.001; **** *p* < 0.0001 (One-way ANOVA followed by the Dunnett post-hoc test).

**Figure 4 ijms-22-02445-f004:**
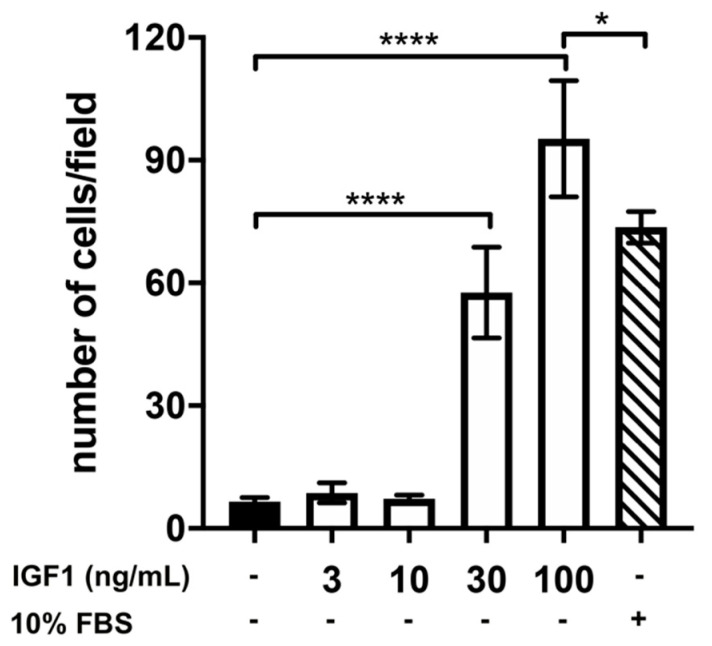
Effect of different concentrations of insulin-like growth factor 1 (IGF1) on GN11 cell migration. Data are expressed as mean ± SD, *n* = 8. One experiment representative of three separate experiments is shown. * *p* < 0.05; **** *p* < 0.0001 (One-way ANOVA followed by the Dunnett post-hoc test).

**Figure 5 ijms-22-02445-f005:**
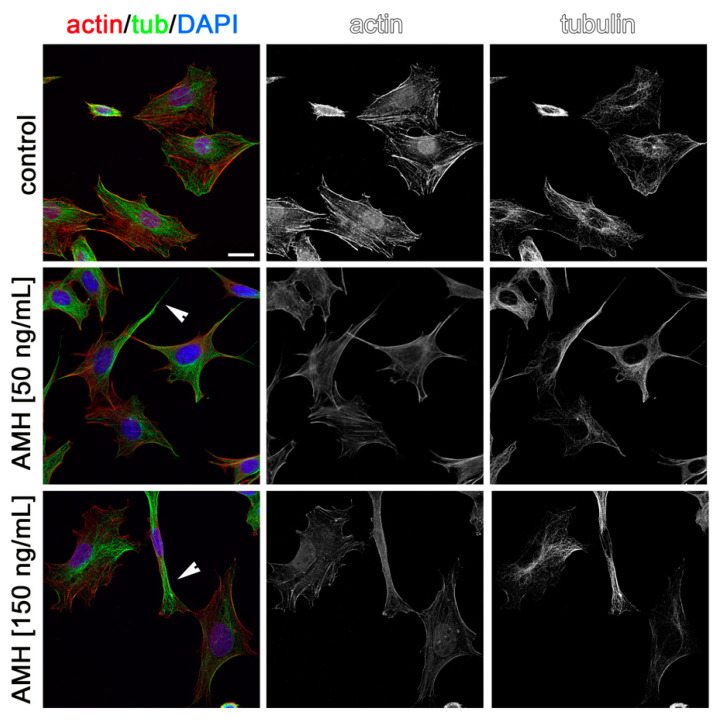
Anti-Müllerian hormone (AMH) treatment alters GN11 cytoskeleton organization. GN11 cells treated with AMH (at 50 and 150 ng/mL) were immune-labeled for F-actin (red) and α-tubulin (green) to reveal microfilament and microtubule organization, respectively. Single channels are shown as black and white images. Arrowheads point at cells displaying a prominent fusiform morphology. Scale bar: 25 µm.

**Figure 6 ijms-22-02445-f006:**
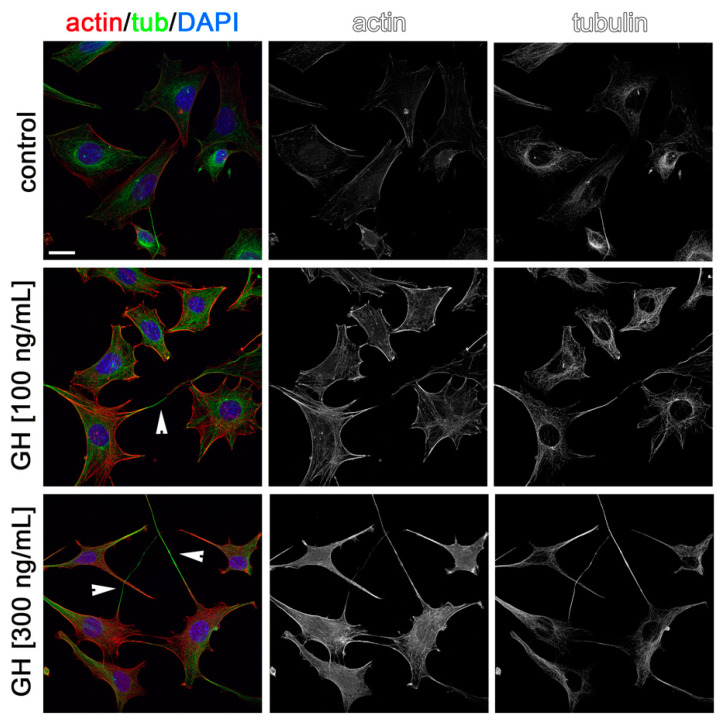
Growth hormone (GH) treatment alters GN11 cytoskeleton organization. GN11 cells treated with GH (at 100 and 300 ng/mL) were immune-labeled for F-actin (red) and α-tubulin (green) to reveal microfilament and microtubule organization, respectively. Single channels are shown as black and white images. Arrowheads point at cells displaying a prominent fusiform morphology. Scale bar: 25 µm.

**Figure 7 ijms-22-02445-f007:**
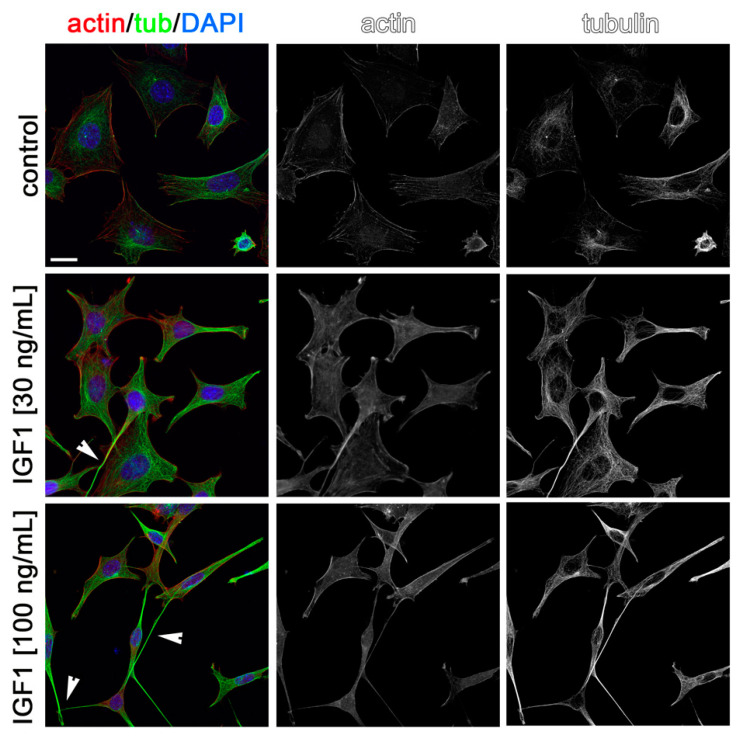
Insulin-like growth factor (IGF1) treatment alters GN11 cytoskeleton organization. GN11 cells treated with IGF1 (at 30 and 100 ng/mL) were immune-labeled for F-actin (red) and α-tubulin (green) to reveal microfilament and microtubule organization, respectively. Single channels are shown as black and white images. Arrowheads point at cells displaying a prominent fusiform morphology. Scale bar: 25 µm.

**Figure 8 ijms-22-02445-f008:**
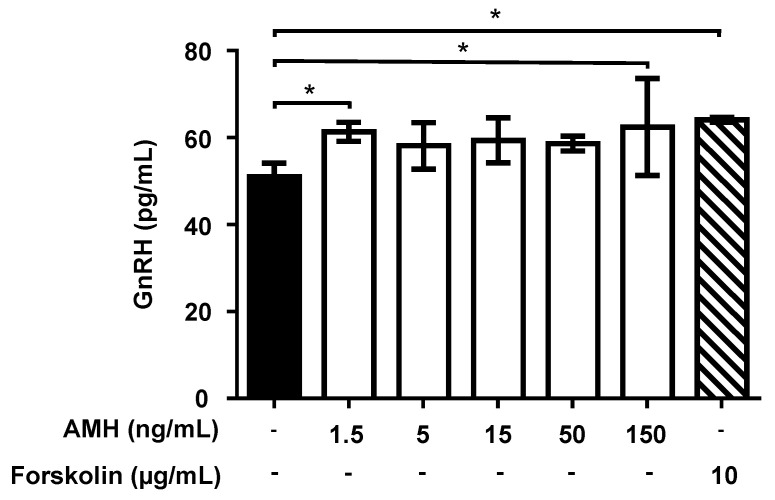
Effect of exposure to Anti-Müllerian hormone (AMH) on gonadotropin-releasing hormone (GnRH) release by GT1-7 cells. Cells were incubated for 90 min with different concentrations of AMH. Data are expressed as mean ± SD, *n* = 4. One experiment representative of three separate experiments is shown. * *p* < 0.05 vs. Control (black bar) (One-way ANOVA followed by the Dunnett post-hoc test).

**Figure 9 ijms-22-02445-f009:**
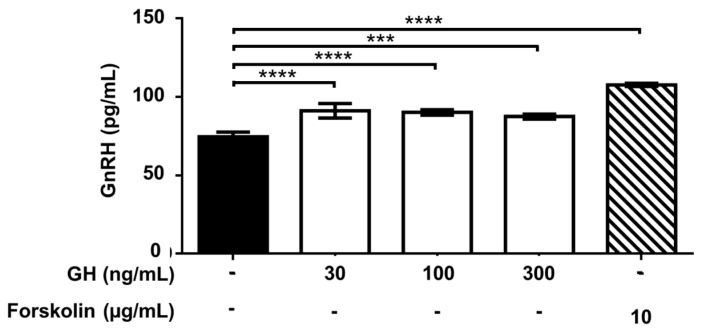
Effect of exposure to growth hormone (GH) on gonadotropin-releasing hormone (GnRH) release by GT1-7 cells. Cells were incubated for 90 min with different concentrations of GH. Data are expressed as mean ± SD, *n* = 4. One experiment representative of three separate experiments is shown. *** *p* < 0.001; **** *p* < 0.0001 vs. Control (black bar) (One-way ANOVA followed by the Dunnett post-hoc test).

**Figure 10 ijms-22-02445-f010:**
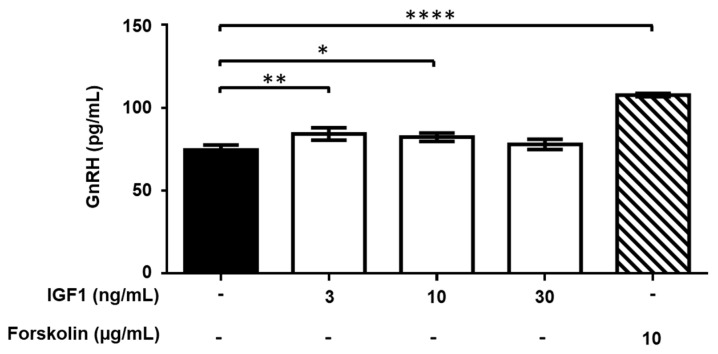
Effect of exposure to insulin-like growth factor 1 (IGF1) on gonadotropin-releasing hormone (GnRH) release by GT1-7 cells. Cells were incubated for 90 min with different concentrations of IGF1. Data are expressed as mean ± SD, *n* = 4. One experiment representative of three separate experiments is shown. * *p* < 0.05; ** *p* < 0.01; **** *p* < 0.0001 vs. Control (black bar) (One-way ANOVA followed by the Dunnett post-hoc test).

**Table 1 ijms-22-02445-t001:** Quantitation (%) of GN11 cells with an elongated morphology after treatment with AMH, GH, and IGF1. AMH, Anti-Müllerian hormone; GH, growth hormone; IGF1, insulin-like growth factor 1; PBS, phosphate buffer saline; HCl, hydrochloride.

Treatment	Percentage of Cells with an Elongated Morphology
PBS	8.3%
PBS HCl	5.9%
AMH (50 ng/mL)	29.2%
AMH (150 ng/mL)	38.6%
GH (100 ng/mL)	36.3%
GH (300 ng/mL)	52.3%
IGF1 (30 ng/mL)	35.8%
IGF1 (100 ng/mL)	48.2%

## Data Availability

The data presented in this study are available on request from the corresponding author.

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
