# Peer review of "Anti-Müllerian Hormone, Growth Hormone, and Insulin-Like Growth Factor 1 Modulate the Migratory and Secretory Patterns of GnRH Neurons"

_ijms, 2021, doi:10.3390/ijms22052445_

Round 1

Reviewer 1 Report

In this study, the authors used immature and mature GnRH neuron cell lines to assess the capacity of several stimuli to affect GnRH cell function and migration. I found the text to be well written and the lines of argument to be well cited. The main limitation of this study is the fact that the authors use only immortalized cell lines to perform their experiments, hence no primary cells are used nor are any in vivo models. My suggestions for improvement are the following:

1. Immunostaining in Figs. 6-7 needs quantification.
2. Because the work presented uses only immortalized cell lines, it is important to perform validation in primary cells.
3. Is chemomigration dependent on AMHR2?
4. The authors should incorporate a discussion of the pros and cons of potential therapeutics that would target this pathway. How specific might they be? What side effects might there be?

Author Response

Reviewer 1

Comments and Suggestions for Authors

In this study, the authors used immature and mature GnRH neuron cell lines to assess the capacity of several stimuli to affect GnRH cell function and migration. I found the text to be well written and the lines of argument to be well cited.

We wish to thanks the reviewer for her/his kind comment.

 The main limitation of this study is the fact that the authors use only immortalized cell lines to perform their experiments, hence no primary cells are used nor are any in vivo models. My suggestions for improvement are the following:

  1. Immunostaining in Figs. 6-7 needs quantification

Reply: as also suggested by the second reviewer, we quantified the percentage of cells with a “migratory” aspect. We have now added these data in the Result section and in the new Table 1.

  1. Because the work presented uses only immortalized cell lines, it is important to perform validation in primary cells.

Reply: We thank the reviewer for his/her comment.

We are aware of the limitation of this study, but as also explained below this is a first study aimed at gathering pilot data that we wish to validate in following studies. Specifically, the aim of this study was to get insights on the possible roles of AMH, GH, and IGF1, in the migration and secretion of GnRH neurons by applying two well established models of GnRH neurons, due to the difficulty in isolating primary cells (GnRH neurons represent a very small population of approximately 1500 cells in rodents). These first observations will be exploited in a second focused study, by employing in vivo models. However, to partially overcome this limitation, we enquired our unpublished data on FAC-sorted primary embryonic GnRH neurons and found the presence of all the three receptors at the three stages analyzed, supporting the physiological importance of these signaling pathways also in vivo. Because these data are still unpublished, we added this information in the Discussion in the last paragraph, as follows:

“Although this study has been primarily performed with immortalized GnRH neuron cell lines, unpublished transcriptomic profiles (available in the Cariboni laboratory) of Fluorescence-Activated Cell-sorted primary GnRH neurons at three developmental stages confirmed the expression of AMHR2, GHR, and IGF1R in GnRH neurons also in vivo.”

  1. Is chemomigration dependent on AMHR2?

Reply: Although we did not perform this control experiment in this study, a previous one (Malone et al., 2019) showed that at least at the dose of 50 ng/mL the effect is mediated by AMRH2, whose activity was abolished by transfecting the cells with a specific shRNA.

  1. The authors should incorporate a discussion of the pros and cons of potential therapeutics that would target this pathway. How specific might they be? What side effects might there be?

Reply: We appreciated this comment. The possible therapeutic insights that come from the results of this study have now been discussed. Briefly, in vivo animal studies should be developed to assess the efficacy of AMH or GH for the treatment of puberty delay or HH. The effects of AMH as therapeutic agent for the treatment of HH or puberty delay has not been explored so far, probably because the scientific evidence of its influence on GnRH neurons has been released only in recent times.

As far as GH, proper therapeutic schemes are to be developed to avoid its negative consequences on the glycolipid metabolism. This has now been included in the last part of the Discussion.

Reviewer 2 Report

This study from Cannarella et al., analyzes the effects of three factors on the migratory potential of GnRH secreting neuron models. Overall the study is well designed with all proper controls and the writing is very clear. There are only a few details and clarifications needed.

In the materials and methods, there is some information missing from the imaging/analysis sections for the migration assays - for example the NA of the objective, why 3 fields only, how many biological replicates (this applies to all the methods), how was the analysis performed?

The same info is missing for studies with the confocal microscope (what objective? how are the images saved etc). while great, the confocal images show a single cell that appears very different - it would make a much better study if the differences in morphology could be quantified or at least define a % of cells in the population with more motile characteristics.  Perhaps live imaging towards a gradient would be the best.

Author Response

This study from Cannarella et al., analyzes the effects of three factors on the migratory potential of GnRH secreting neuron models. Overall the study is well designed with all proper controls and the writing is very clear.

Thanks for this comment.

There are only a few details and clarifications needed.

In the materials and methods, there is some information missing from the imaging/analysis sections for the migration assays - for example the NA of the objective, why 3 fields only, how many biological replicates (this applies to all the methods), how was the analysis performed?

Reply: We added this information in the Methods in the section “2.6. Cytoskeleton labeling”, as follows: “Preparations were then examined with an epifluorescent fluorescent microscope (Zeiss, Milano, Italy), and images were acquired with a Zeiss LSM900 laser scanning confocal microscope and a 40X objective (Zeiss Plan-Apochromat 40X, NA 1.3, Oil-immersion). ZEN 3.0 software (Zeiss) was used to process z-stacks at 0.25 µm intervals and generate maximal intensity projection images. Quantification of the percentage of the cells with an “elongated” shape was performed by counting 5 fields/conditions by ImageJ.”

The same info is missing for studies with the confocal microscope (what objective? how are the images saved etc). while great, the confocal images show a single cell that appears very different - it would make a much better study if the differences in morphology could be quantified or at least define a % of cells in the population with more motile characteristics. 

Perhaps live imaging towards a gradient would be the best.

Reply: It would have been very interesting to perform ‘in vivo’ imaging, but the laboratory space restrictions at the dedicated imaging facility equipped for live-imaging have hampered us to perform these assays. Thus, to overcome this problem, we quantified the percentage of cells with elongated shape upon the different treatments. This information is now presented in new Table 1.